# Interphase chromosomes of the *Aedes aegypti* mosquito are liquid crystalline and can sense mechanical cues

Vinícius G. Contessoto [1,2] ✉, Olga Dudchenko [1,3], Erez Lieberman Aiden[1,3], Peter G. Wolynes [1,4,5,6], José N. Onuchic [1,4,5,6] ✉ & Michele Di Pierro [7,8] ✉

We use data-driven physical simulations to study the three-dimensional architecture of the *Aedes aegypti* genome. Hi-C maps exhibit both a broad diagonal and compartmentalization with telomeres and centromeres clustering together. Physical modeling reveals that these observations correspond to an ensemble of 3D chromosomal structures that are folded over and partially condensed. Clustering of the centromeres and telomeres near the nuclear lamina appears to be a necessary condition for the formation of the observed structures. Further analysis of the mechanical properties of the genome reveals that the chromosomes of *Aedes aegypti*, by virtue of their atypical structural organization, are highly sensitive to the deformation of the nuclei. This last finding provides a possible physical mechanism linking mechanical cues to gene regulation.

Within the nucleus, the genome of eukaryotes folds into partially organized three-dimensional structures specific to the cell type and phase of life. It is increasingly evident that these architectural features of the genome are related to the process of transcriptional regulation; the disruption of such organization has been observed to lead to disease[1–6]. In interphase, chromatin architecture brings together sections of DNA separated by long genomic distance and modulates the interactions between genes and regulatory elements. Additionally, the position of genes within nuclei appears to correlate with transcription, with repressed or heterochromatic regions being often found in the vicinity of the nuclear lamina[7] while transcriptionally active loci are preferentially positioned in the outer portion of chromosomal territories[8–11].

In the past decade, DNA-DNA proximity ligation assays have opened the way to the systematic study of genome architecture. The high-throughput ligation assay called Hi-C reports the frequency of chromatin contacts genome-wide[12–17]. Hi-C assays have shown that chromatin segregates into regions of preferential long-range interactions referred to as compartments[16,17]. In general, one group of compartments, the A compartments, tends to be more gene-rich and contains a more significant fraction of active chromatin than do the B compartments. Theoretical models have shown that a process akin to liquid–liquid phase separation is likely to be responsible for the segregation of chromatin characterized by similar epigenetic marking patterns into distinct genomic compartments[8,18–20].

At a smaller scale (tens to hundreds of kilobases), DNA-DNA ligation assays have shown the existence in a wide range of organisms of chromatin domains that are characterized by preferential internal interactions, commonly referred to as "topologically associating domains" (TADs)[21–26]. In mammals, the boundaries of TADs are often characterized by the presence of CCCTC-binding factor (CTCF) and structural maintenance of chromosomes (SMC) protein complexes[17]. In the presence of both of these factors, the two loci bordering a TAD form strong interactions, which are visible in Hi-C maps as peaks in

[1]Center for Theoretical Biological Physics, Rice University, Houston, TX, USA. [2]Instituto de Biociências, Letras e Ciências Exatas, UNESP - Univ. Estadual Paulista, Departamento de Física, São José do Rio Preto, SP, Brazil. [3]The Center for Genome Architecture, Department of Molecular and Human Genetics, Baylor College of Medicine, Houston, TX, USA. [4]Department of Physics & Astronomy, Rice University, Houston, TX, USA. [5]Department of Chemistry, Rice University, Houston, TX, USA. [6]Department of Biosciences, Rice University, Houston, TX, USA. [7]Department of Physics, Northeastern University, Boston, MA, USA. [8]Center for Theoretical Biological Physics, Northeastern University, Boston, MA, USA. ✉e-mail: vinicius.contessoto@rice.edu; jonuchic@rice.edu; dipierro@northeastern.edu

contact frequency. Notably, at the TAD boundaries, CTCF binding motifs are almost always seen in convergent orientation.

To explain this last feature, it was proposed that DNA loops are extruded by SMC complexes; a phenomenon subsequently confirmed to occur by in vitro live imaging[27]. According to the "loop extrusion model", the process of DNA extrusion is halted by the presence of a pair of CTCF proteins bound to the polymer with convergent orientation[28–30]. As a consequence, while DNA extrusion is thought to be ubiquitous along chromosomes, peaks in contact probability are predicted to be formed only at the anchors of a loop domain, as observed with Hi-C. The "loop extrusion model" has since been backed-up by numerous studies that involve the depletion in vivo of cohesin, the cohesin-loading factor Nipbl, or CTCF[31–35].

The observations just summarized, together with several theoretical studies[8,18,36–40], demonstrate that chromosomal architecture is shaped by the two processes of lengthwise compaction, driven by motor activity (which gives rise to the Ideal Chromosome model[8,41,42]), and phase separation, which is controlled by epigenetic marking patterns[18]. While the physical mechanisms and the molecular machinery behind the formation of genome architecture seem to be largely shared among organisms, the resulting genome architectures are far from unique. Ongoing efforts characterizing the genomic structural ensembles of many species have found an assortment of distinct chromosomal spatial organizations[43–46]. How the two processes of phase separation and lengthwise compaction generate this collection of shapes remains an open question.

Here, we study the chromosomal architecture of the mosquito *Aedes aegypti* - a vector transmitting the viruses responsible for several tropical fevers, including dengue, chikungunya, zika, mayaro, and yellow fever[47–52]. *Aedes aegypti*'s genome is roughly half the length of the human genome and is organized only into 3 chromosomes; thus leading to comparably large chromosomes[53]. Furthermore, Hi-C assays reveal the existence of preferential interactions among chromosomal telomeres and among the centromeres. Conversely, the telomeres rarely make contact with the centromeres. This pattern indicates that the centromeres and telomeres are separated from each other, and the resulting nuclei are polarized. These observations suggest that the chromosomes are arranged similarly to the so-called Rabl configuration[54], a type of genome architecture previously seen in many eukaryotes of variable genome size, including yeast and plants[43,55–57]. In this polarized configuration, chromosomes are folded over into a hairpin-like structure that persists over time, with the centromeres and the telomeres possibly anchored to the nuclear lamina[55,58]. While it has been suggested that the Rabl configuration may help in reducing chromatin entanglement[59,60], it remains unclear whether organisms exhibiting different types of genome architecture display different entanglement levels. It has been reported that the absence of condensin II subunits correlates with different genomic organization types in several species. Also, theoretical models suggest that the lengthwise compaction associated with motor activity counteracts phase separation[43].

In this paper, we use a data-driven physical simulation to decode the information contained in the Hi-C maps of *Aedes aegypti*. We use a polymer model for chromatin that includes phase separation driving the compartmentalization of chromatin as well as the effects of motor activity, as described by the Ideal Chromosome potential[8]. We also show that polarization, *i.e.* the clustering of telomeres and centromeres at opposite ends of the nucleus, is a necessary condition for the formation of the observed structures. One possible mechanism of polarization, which we employ, is through anchoring to the nuclear envelope. While other polarization mechanisms are possible, the analysis of the structural ensembles of the chromosomes would remain largely unchanged.

At first glance, one would think that if two mechanisms existed, one giving rise to compartmentalization and another giving rise to polarization, the result of their concurring activity would be partial compartmentalization. A key finding of our investigation is that this is not so; in fact, partial compartmentalization requires not only fold-back, but also enhanced short-range extrusion during interphase.

By optimizing the parameters in our physical model to produce an ensemble of 3D structures that is in optimal agreement with the DNA-DNA proximity ligation maps of *Aedes aegypti*, we show that the experimental crosslinking data themselves imply that *Aedes*'s interphase chromosomes are subject to a greater amount of short-range lengthwise compaction than what has been observed in other organisms[43].

The combined effects of the polarized configuration and enhanced short-range extrusion create unusually elongated chromosomal territories, such that individual chromosomes occupy non-overlapping regions of space but that nevertheless interact significantly at the surface of those regions. In such a configuration, genomic contacts are formed only within relatively short contiguous chromatin segments or across the territorial boundaries with opposite-facing loci. This explains why the Hi-C maps exhibit a broad main diagonal and an enhanced probability of contacts along a secondary diagonal. Compartmentalization is observed along the two diagonals, indicating that chromatin in *Aedes aegypti*, while partially condensed by strong short-range lengthwise compaction, remains fluid and can rearrange to accommodate phase separation[41]. It is worthwhile mentioning that liquid crystalline structures have also been suggested as the genomic architecture for some systems[61–63].

Finally, we explore the effect of deforming the shape of the nucleus on the formation of contact interactions within the mosquito's genome. In stark contrast to the relative insensitivity of territorialized chromosomes to deformation - mammalian chromosomes, for example - we find that changing the shape of the nuclei of *Aedes aegypti* results in dramatic changes in the contact patterns of the Rabl configuration. Besides the obvious influence of the anchoring of centromeres and telomeres to the lamina, we find that the high level of lengthwise compaction found in *Aedes*'s interphase chromosomes leads to high sensitivity of its genomic contact patterns to mechanical cues. This last finding constitutes an intriguing feature of the Rabl configuration and suggests a possible physical mechanism linking mechanical signals on the cell to gene regulation.

## Results

### 3D modeling reveals that interphase chromosomes in *Aedes aegypti* are partially condensed

The first task at hand is to calculate the ensemble of 3D chromosomal structures that best corresponds to the observed Hi-C maps. This can be done using the Maximum Entropy (MaxEnt) principle and polymer physics[8,41]. One such model, called the Minimal Chromatin Model (MiChroM) has already been employed with success in investigating chromosomal organization in interphase in many organisms[8,9,18,19,43,64,65]. To obtain the interaction parameters specific for the *Aedes aegypti* genome, the MiChroM energy function was re-trained using the Hi-C data for the *Aedes aegypti* chromosome 1[45] with a resolution of 100 kilobases. A/B compartment annotations were also obtained from the Hi-C maps using the first eigenvector of the Hi-C correlation matrix[16]. Additionally, we incorporated harmonic restraints to reproduce the clustering of telomeres and centromeres and to anchor telomeric regions to one side of the nucleus wall and centromeres to the opposite side[55,58] (see "Methods" for details). The newly trained MiChroM model is then used for the chromatin simulations using Langevin dynamics[66] in order to produce an ensemble of hundreds of thousands of 3D structures for the chromosomes of *Aedes aegypti*. To verify that this ensemble of genomic conformations does indeed represent what is found in the DNA-DNA ligation assays, the experimental Hi-C maps (Fig. 1A) are compared with the Hi-C maps predicted by the in silico ensemble (Fig. 1B). The similarity of the two sets of Hi-C maps is

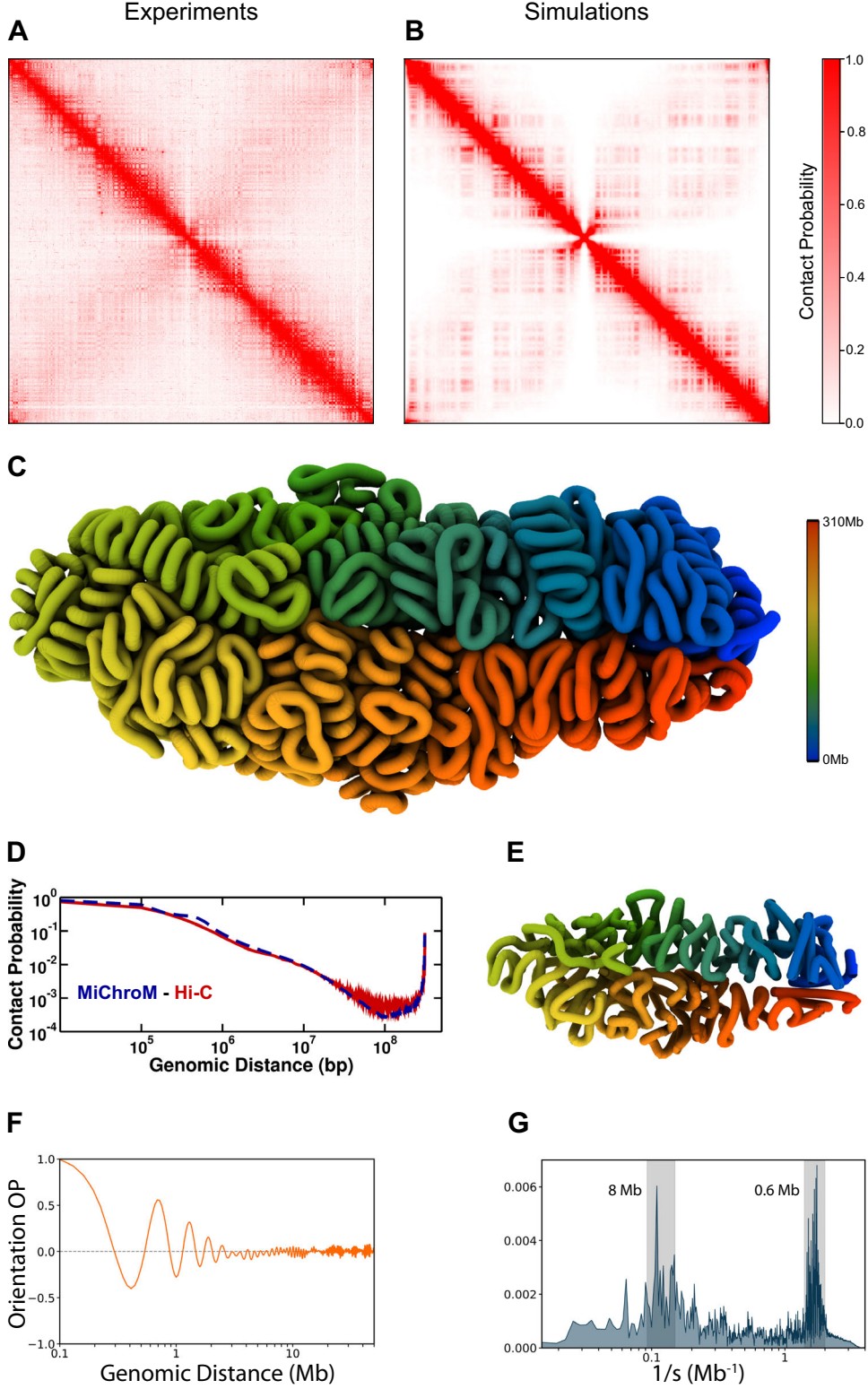

**Fig. 1 | Hi-C maps and 3D structures of chromosome 1 after the parameter's minimization. A** Hi-C map of chromosome one from experiments[45,74]. **B** In silico Hi-C map obtained by the ensemble of 3D structures generated by simulations. The red color intensity is related to the probability of a loci contact formation. Pearsons' correlation between the experimental and the in silico map is 0.957 (see Supplementary Fig. 7B). **C** 3D quenched representative structure of chromosome 1 at 100 kb resolution. The color scheme paints the locus by their sequential position from blue to red, head to tail, respectively. **D** Contact probability as a function of genomic distance. In red is the curve obtained from the experimental Hi-C. The dashed blue line represents the polymer scaling obtained from the in silico Hi-C map. **E** Coarsened structure of chromosome 1 from the quenched structure presented in (**C**). The segment representation is an average of ten loci in the polymer chain. **F**, **G** The orientation order parameter ($O_{OP}$) as a function of the genomic distance and its Fourier transform, respectively.

immediately evident; the Pearsons' correlation between the simulation and the experiments is $R = 0.957$ (Pearson's correlation as a function of the genomic distance and the overall correlation are shown in Supplementary Fig. 3 and Supplementary Fig. 7). As previously mentioned, both sets of Hi-C maps show very distinct properties when compared to those observed in most mammalian cell lines. The main diagonal appears particularly wide, indicating the frequent formation of contacts between loci further away in genomic distance than what is commonly seen in mammals. In our recent work[43], we quantified these properties by calculating the ACA (Aggregate Chromosome Analysis) on in situ Hi-C maps of 24 species. The architectural features observed in those maps can be divided into two groups, type-I (Rabl-like configurations such as centromere clustering, telomere clustering, and telomere-to-centromere axis) and type-II (chromosome territories). The details of the ACA score and the comparison of the contact probability curves for the human and mosquito chromosome 1 are presented in the Supplementary Information (Supplementary Fig. 4). Figure 1D presents the contact probability as a function of the genomic separation and shows that the compaction seen in the polymer model is consistent with the experimental data. A similar wide diagonal, corresponding to a high degree of compaction, has been observed in the Hi-C maps of mitotic chromosomes[67–69]. In addition, a secondary diagonal is observed in the maps; besides these two diagonals, the frequency of intra-chromosomal contacts is greatly depleted with respect to what is seen in mammalian cells. Analyzing the ensemble of 3D structures associated with the Hi-C maps, we see that these features correspond to partially condensed chromosomal conformations. These conformations do indeed visually resemble those of chromatids (Fig. 1C). This is in stark contrast with the roughly globular chromosomal conformations typically observed in interphase in mammals[8,19,40,43,65,70–73]. In addition, the *Aedes aegypti* chromosomes are folded over, generating frequent interactions between the two arms, which are manifested as the secondary diagonal (Coarsened 3D structure - Fig. 1E). Crucially, we find that both polarization and the shortening of the chromosomes are needed for the emergence of this doubled-over architecture, reflected in the features observed in the Hi-C maps. In the MiChroM energy landscape, the shortening of the chromosomes is due to a high degree of lengthwise compaction, a term related to the ideal chromosome potential that we have shown to be directly related to the activity of SMC complexes[43]. This last observation seems to suggest that the partial condensation observed in the chromosomes of *Aedes aegypti* is due to increased DNA extrusion by SMC complexes. The question is however more complicated, because, while in the mosquito's chromosomes short-range genomic contacts are enhanced, long-range contacts are depleted; so, differential extrusion appears a more likely explanation rather than increased extrusion per se[43,45,74,75]. The origin of the specific lengthwise compaction profile characteristic of *Aedes aegypti* is beyond the scope of this study. We expect one organism's profile to depend on a variety of different known and unknown factors; among them, the individual concentrations of the many different SMC complexes as well as the presence of CTCF, which is likely to interfere with cohesin extrusion. Additional insights about the genomic energy landscape of the mosquito can be obtained through a process of a quenching simulation; in this way, the natural structural tendencies of the chromosomes are uncovered by removing thermal disorder. Quenched chromosomes appear to adopt conformations composed of helices of helices, another remnant of mitotic chromosomes[67–69,76]. The tendency of chromatin to form hierarchical helices has been observed in human chromosomes as well and is perhaps a universal feature of eukaryotic genomes[41]. Representative 3D structures of chromosome 1 at the nominal information theoretic temperature ($T = 1.0$) are presented in Supplementary Fig. 1.

As mentioned before, such genome architecture resembles that of liquid crystals (partially ordered). The chromatin fiber of *Aedes aegypti* flows and changes shapes similar to a human chromosome in interphase that is described as being liquid-like. This first aspect, like liquid droplets, allows for A/B phase separation and compartmentalization (observed in the anti-diagonal in Fig. 1A, B). However, there is local structural order in the form of helices that leads to an orientational order along the genomic distance that is similar to what is seen for the mitotic chromosome - thicker diagonal in Hi-C maps of Fig. 1A, B). To quantify such structural property, we also calculated an orientation order parameter ($O_{OP}$) previously employed to investigate local rearrangements in mitotic chromosomes[41]. The parameter $O_{OP}$ is defined as the correlation between two unit vectors connecting beads $[i, i + 4]$ and $[j, j + 4]$ (see the Supplementary Information for details). The analyses were performed for the quenched structure and the ensemble of 3D structures at $T = 1.0$ (see Supplementary Fig. 1E, F for details). Figure 1F shows that $O_{OP}$ oscillates as a function of the genomic separation that can be associated with fibril structures[41]. The Fourier transform of $O_{OP}$ presents two regions with intense values of the spectrum, indicating that there are two layers of fibril structures as shown in Fig. 1G (see Supplementary Fig. 2 for human). The first is related to a higher turn frequency with a periodicity around 0.6 Mb which can be observed as local helicoidal structures shown in Fig. 1C. In addition, the second layer has a periodicity of longer genomic separation in the range of 8 Mb.

## Elongated territories lead to local compartmentalization and extensive inter-chromosomal interactions

MiChroM interaction parameters for *Aedes aegypti* determined from chromosome 1 were then used to simulate its complete genome, consisting of both the maternal and paternal copies of each of the three chromosomes typically present in the *Aedes aegypti* family (see the Supplementary Information for details). The inter-chromosomal Hi-C maps generated in silico, once again, were very similar to those found experimentally, as shown in Fig. 2A; in particular, it is evident that the high contact frequency between the telomeres and centromeres of different chromosomes[45] is very well reproduced. The whole nucleus simulation of the mosquito's genome allows us to examine territorialization, compartmentalization, and inter-chromosomal interactions. Chromosomal territories are non-overlapping spatial domains occupied by a single chromosome, typically characterized by a globular shape; such shape increases the frequency of intra-chromosomal contacts and decreases the frequency of inter-chromosomal contacts. Territories are easily visible in Fluorescence In-Situ Hybridization (FISH) images and Hi-C maps of most mammals[16,77]. According to this definition, territories are not present in *Aedes aegypti*[43,45]. More careful examination of the ensemble of 3D structures shows that non-overlapping spatial domains occupied by a single chromosome do exist, but display a very elongated shape. Furthermore, the unusually asymmetric shape of the *Aedes aegypti* territories, by virtue of their high surface-to-volume ratio, does not significantly limit the frequency of inter-chromosomal interactions; yet, chromosomes remain spatially well separated and are untangled (Fig. 2).

Next, we examined compartmentalization. As mentioned previously, the wide main diagonal of *Aedes*'s intra-chromosomal ligation maps resembles that of mitotic chromosome maps[41,68]. In contrast with the mitotic case, however, *Aedes aegypti* chromosomes show compartmentalization along the diagonal - a characteristic commonly also found in the human interphase but that usually extends to the whole map. Similarly, compartmentalization is apparent along the secondary diagonal. The presence of compartmentalization indicates that the chromosomes, while being condensed, are still fluid enough to accommodate local phase separation. Chromatin micro-droplets are formed among small contiguous segments of one arm (main diagonal) or are comprised of loci on opposite arms but which have been brought into proximity by the folded over configuration (secondary diagonal) (Fig. 2). The epigenetically driven phase separation driving

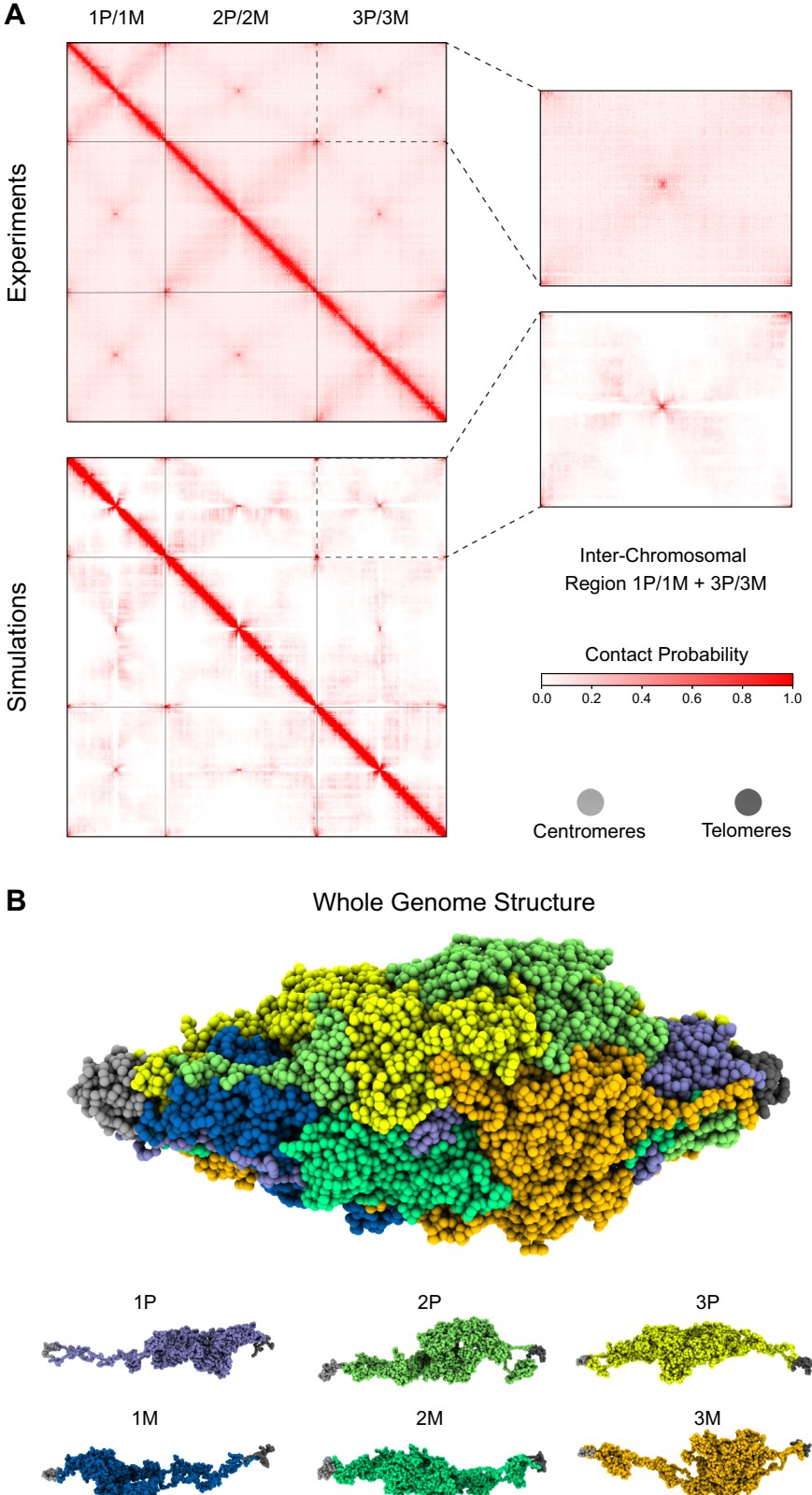

**Fig. 2 | Full nucleus simulation of the *Aedes aegypti* genome. A** Experimental (top) and in silico (bottom) Hi-C maps of the *Aedes aegypti* genome. The zoom-in region shows the inter-chromosomal interactions between chromosomes 1 and 3. The whole genome Hi-C maps emphasize the centromere-centromere and telomeres-telomeres preferential contact formation. **B** Full nucleus representative 3D structure. Each chromosome chain is represented by a unique color. Centromeres are represented by ligth gray beads (left) and telomeres by dark gray (right). The 3D structure highlights the formation of elongated territories in the *Aedes aegypti* genome.

compartmentalization competes with the ordering of the helical fibers, in which defects are formed to accommodate the creation of micro-compartments[43]. The coexistence of both ordering and fluidity indicates that the *Aedes aegypti* chromatin manifests properties typical of a liquid crystal[41,78].

Somatic pairing of homologous chromosomes is a phenomenon that could play a role in the 3D organization of chromosomal territories in *Aedes*. Pairing between the two homologous chromosomes has been observed in a variety of species belonging to the Diptera order[43,79–81]. While at the moment direct evidence of pairing in *Aedes aegypti* is still lacking, we nevertheless expect *Aedes* to exhibit some degree of pairing. In absence of suitable experimental data, we decided not to include any mechanism of pairing in our physical modeling. We do not expect pairing of homologous chromosomes to significantly changes the shape of the *Aedes aegypti* chromosomal territories or their internal organization. However, pairing may have an effect on the positioning of chromosomes within nuclei and in compartmentalization acting across territories, which for the moment remains unaccounted. Additionally, it remains unclear if the homologous pairing is a phenomenon that is functionally related to the partially condensed chromosomes observed in this manuscript or to the Rabl configuration, or if these are just uncorrelated contingencies.

In human cells, active chromatin tends to be positioned at the periphery of chromosomal territories[8,9] and thus frequently interacts with other chromosomes[11]. We study the spatial distribution of active chromatin using, previously published, ATAC-seq (Assay for Transposase-Accessible Chromatin using sequencing) data for *Aedes aegypti* brain.[74,82]. The ATAC-seq signal identifies accessible DNA regions and it has been reported to correlate positively with their belonging to A compartments[83,84]. For *Aedes aegypti*, we do indeed find a mild correlation, with the high-intensity values of the ATAC-seq signal belonging in the majority (64%) to the A compartments, while the B compartments contain most of the low-intensity signal (59%) values (see Supplementary Fig. 6 and the Supplementary Information for details). The low correlation is possibly due to the fact that Hi-C experiments were performed using whole body extract, while the ATAC-seq experiments were performed using only brain cells. Using the 3D genomic structural ensemble of the mosquito, we measure the radial positioning of the ATAC-seq peaks—a proxy for high activity—with respect to the axis of the chromosomes as defined by the line joining telomeres and centromere. Similar to what was observed in human cells, we find that active chromatin is preferentially located in the outer shell of *Aedes aegypti*'s elongated territories[85,86] Fig. 3. Further support to this finding comes from quantifying how often each locus is exposed to the surface of the territory, and therefore available to form inter-chromosomal contacts (Fig. 3). Overall, just as in humans, a disproportionately large fraction of inter-chromosomal interactions occurs among active chromatin regions. The preservation of this feature of genome organization across species and architectural types[43] suggests that the positioning of active chromatin at territorial interfaces might have a role in gene regulation[85,86].

Nevertheless, significant differences between the chromosomal architecture of mammals and *Aedes* mosquitoes are seen even in the case of active chromatin positioning. In mammals, we have observed that the chromatin segments belonging to the same genomic compartment—and thus carrying similar epigenetic markings—form liquid droplets. In three dimensions, these droplets rearrange dynamically by splitting and fusing, leading to the emergence of genome-wide compartments observed in DNA-DNA ligation assays[8,18,64]. In *Aedes aegypti*, similarly to what one finds for human chromosomes, the active chromatin forms droplets (A/B micro phase-separation); but, due to the increased condensation and the elongated shape of the territories, these droplets are less likely to fuse with similar droplets situated at distant positions along the chromosomal axis. Thus, the global structure

leads to the formation of only local compartments. In contrast with what is seen in mammals, in *Aedes aegypti* the loci diffuse only within local compartments but do not mix with far away chromatin even when the distant chromatin carries similar epigenetic markings to a given local compartment, i.e., the Rabl-like genome architecture of mosquitoes, causes the contacts between loci that are located near the polarized regions (Centromere and Telomeres) to form less frequently. The distinct nature of compartments in *Aedes* is likely to have some repercussions on transcriptional regulation.

## Chromosomal structures of *Aedes aegypti* are sensitive to mechanical cues

As discussed, our model indicates that the anchoring to the nuclear envelope of the centromeres and telomeres is necessary for the formation of the genomic architecture observed in *Aedes aegypti*. This spatial connection between the chromosomes with the nuclear envelope might play a role in transducing mechanical cues into gene regulation. To investigate this intriguing possibility, we studied the effect of deforming the shape of the nucleus on the ensemble of conformations of the mosquito's genome (Fig. 4—left panels). We apply tension and compression along the axis joining the telomeres and centromeres; in both cases, the resulting deformation is set to 30% of the initial distance. Compression reduces the asymmetry in the shape of territories. In this case, the in silico Hi-C maps display less prominent primary and secondary diagonals with, conversely, an increased frequency of promiscuous intra-chromosomal contacts distributed along the entire chromosome is found. In other words, under compression, the mosquito's chromosomes resemble more mammalian chromosomes than they do without compression. Likewise, stretching the nucleus results in an opposite effect, with more prominent primary and secondary diagonals and generalized depletion of all other intra-chromosomal interactions. Overall, it is clear that in *Aedes aegypti* deformations of the nuclear shape are able to shift the ratio between intra-chromosomal and inter-chromosomal interactions. Moreover, large shape deformations can make frequent genomic contacts otherwise infrequent, and vice versa. This modulation of contact frequencies that would result from nuclear deformation could provide a physical mechanism by which mechanical cues could influence transcriptional regulation. The present model indicates that the sensitivity of the Rabl architectural type toward changes in the shape of the nucleus is due to both anchoring of the chromosome to the nuclear envelope and increased lengthwise compaction. Mammalian chromosomes, in contrast, appear relatively insensitive to changes in the shape of the nucleus, with their Hi-C maps hardly showing any differences in the contact frequencies upon shape deformations. To establish whether anchoring alone could be responsible for the increased mechanical sensitivity of the mosquito's genome, we simulated the mosquito chromosome with MiChroM interaction parameters tuned to reproduce the human Hi-C maps (Fig. 4—right panels). With this tuning, because of the lesser degree of lengthwise compaction, only very minor changes in the Hi-C maps were found when applying longitudinal deformations. It is suggestive that the very same features leading to the emergence of the Rabl configuration are also responsible for its sensitivity to mechanical signals. Notwithstanding the challenge of finding a relationship between crosslinking frequencies with physical-spatial distances[37,72,73,87–91], we also observed a good agreement of the chromosome length with experiments when comparing it with other studies. In our previous works, we employed a bead diameter of $\sigma_{50kb} = 0.165$ μm using the MiChroM energy function at 50kb resolution. Here we used a locus resolution of 100 kb per model bead. Assuming a constant density of

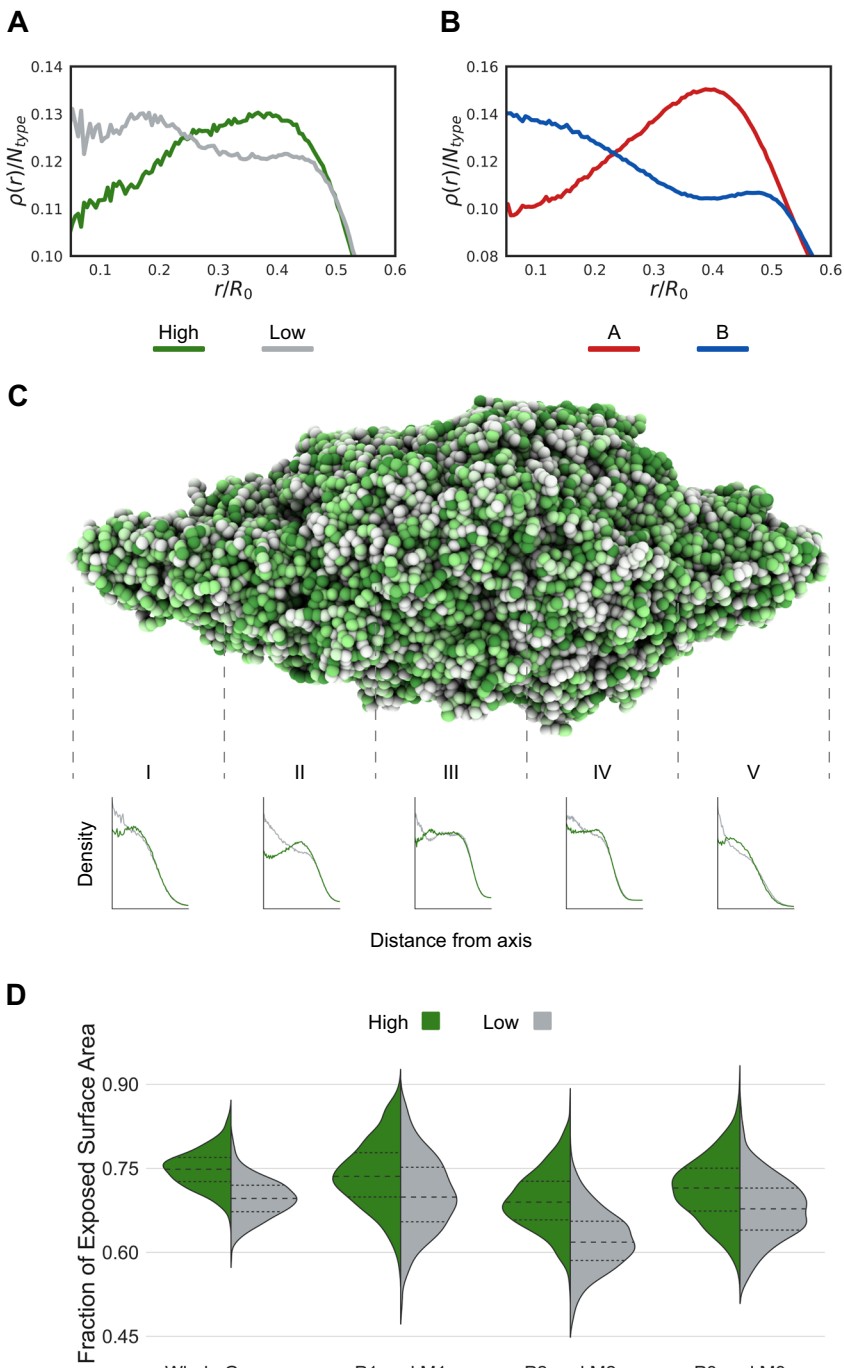

**Fig. 3 | Loci characterized by high-intensity ATAC-seq signals are often located on the chromosome surface. A** Radial density distribution for low and high ATAC-seq signal intensity (gray and green lines). **B** Radial density distribution for chromatin types A and B (red and blue lines). Analogous to High ATAC-seq signals, chromatin type A is often presented in the chromosome periphery.
**C** Representative 3D structure of the *Aedes aegypti* full nucleus colored based on ATAC-seq signal. Cylindrical density distributions are presented for five sections along the centromere-telomeres axis. **D** Violin plots show the normalized exposed area for beads with low and high ATAC-seq signal intensity. The distributions are shown for the whole genome and for each chromosome. Higher values mean the locus is exposed and more often located at the periphery (interacting with the neighbors chromosome).

chromatin between different bead resolutions, the bead diameter for the 100 kb resolution is $\sigma_{100\,kb} = \sqrt[3]{2}\sigma_{50\,kb} = 0.20789\,\mu m$. This value leads to the estimate of the centromere-telomeres distance (which corresponds to the nucleus diameter $L_0 = 38\sigma_{100\,kb}$) around 7.9 µm. The total length of Aedes aegypti chromosome 1 in interphase is approximately 15.3 µm. It is reported in the literature that chromosome 1 in the earlier stage of mitosis has an average length of 11.86µm[92]. This is a quite good agreement given

that there should be some condensation during the initial stages of mitosis which is related to chromosome shortening.

## Discussion
The explosion of activity to characterize genome architecture across the tree of life is revealing that genomic structural ensembles are far more diverse than previously believed[43]. Yet, it is reasonable to assume that the physical processes that form this architectural diversity should

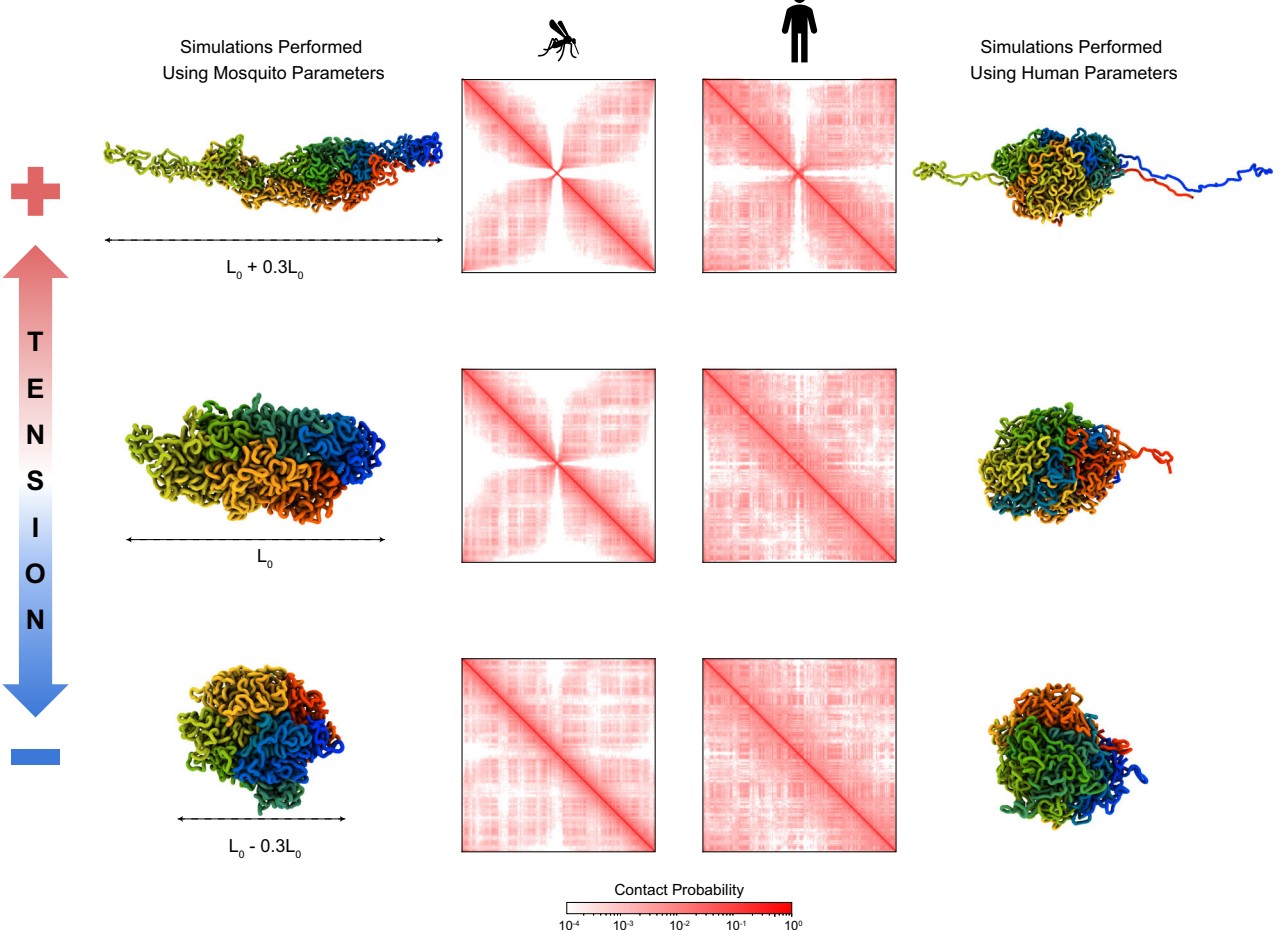

**Fig. 4 | Effects of changing the centromere-telomeres separation distance.** Left panels show the results applying more and less tension in the mosquito genome. Right panels present the analysis of simulations using the Human genome parameters (MiChroM[8]). Representative 3D structures of each condition are shown alongside the in silico Hi-C maps. The top panels present the centromere-telomeres separation distance $L_+$, 30% bigger than $L_0$ that is presented in the middle. At the bottom is present the case where the separation distance $L_-$ is 30% smaller than $L_0$.

be similar because the emergence of new physical processes is ultimately limited by the slow process of molecular evolution. Our analysis shows that a universal physical model can account for two very different genome architectural types; the territorialized architecture of mammals and the polarized architecture of *Aedes* mosquitoes. Our physical model consists of a simple polymer subject to the two processes of motor-driven lengthwise compaction and epigenetically driven phase separation; both of these processes have been supported by numerous studies in vitro and in vivo. Besides the origin of genome architectural types, a fundamental question in physical genetics pertains to the functioning of the different architectural types. The length of an individual chromosome is likely to play a role in determining the optimal genome architecture. It is possible that the increased short-range lengthwise compaction seen in *Aedes aegypti* is an adaptive mechanism to accommodate the large chromosomes of the mosquito. On the other hand, increased lengthwise compaction also leads to high sensitivity with respect to mechanical cues, so, perhaps, the Rabl configuration evolved as a sensing apparatus. Regardless of the evolutionary trajectory, combining theoretical and computational analysis with experiments is essential toward unraveling the mechanisms by which 3D architecture influences the functioning of genomes.

## Methods
The genome of the mosquito *Aedes aegypti* was modeled using a polymer physics model where parameters are found using the

maximum entropy principle. The resulting model resembles the Minimal Chromatin Model (MiChroM) that was successfully employed to investigate the genome organization for human chromosomes in different cell lines in interphase[8,9,18,19,64,65]. The MiChroM energy function has two assumptions; The first presumes that the chromosome phase separation is related to the A/B chromatin types annotation. The second considers the different proteins' motor activity to be related to the polymer compaction through an ideal chromosome interaction. The MiChroM potential has the following form:

$$U_{\mathrm{MiChroM}}(\vec{r}) = U_{HP}(\vec{r}) + \sum_{\substack{k \geq l \\ k,l \in \text{Types}}} \alpha_{kl} \sum_{\substack{i \in \{\text{Loci of Type k}\} \\ j \in \{\text{Loci of Type l}\}}} f(r_{ij}) + \sum_{d=3}^{d_{cutoff}} \gamma(d) \sum_i f(r_{i,i+d})$$

$$(1)$$

where $U_{HP}(\vec{r})$ is the potential energy of a generic homopolymer (see the Supplementary Methods section "Homopolymer Model" for details); $\alpha_{kl}$ is the strength of the energy interactions between the chromatin type $k$ and $l$; $\gamma$ is the ideal chromosome energy interaction as a function of the genomic distance $d$, and the $f(r_{ij})$ represents the probability of crosslinking of loci $i$ and $j$ (see the Supplementary Methods section "Crosslinking Probability Function" for details). Similar to the approach taken for the human genome investigation[8], here it was also necessary to obtain the interaction energy

parameters for each term of the potential $\alpha$ and $\gamma$. The model was trained using the data for chromosome 1 of the *Aedes aegypti* obtained from the Hi-C map[45,74] with a resolution of 100 kb per bead. The chromatin types annotation A and B were obtained from the first eigenvector from the experimental data correlation matrix[16]. An additional assumption was included during the training of the potential parameters by incorporating spatial position restraints for the chromosome telomeres and the centromere. Telomeric regions are positioned on one side of the nucleus wall and the centromere on the opposite side. This assumption is based on possible interactions of these chromosome regions with lamins that are found in the nuclear envelope[55,58]. Chromatin dynamics simulations were performed using Gromacs package version 2016.3[93]. The simulations employed Langevin dynamics and followed the protocol described in the Nucleome Data Bank (NDB) and Open-MiChroM software documentation (https://open-michrom.readthedocs.io)[9,19]. The parameters minimization $\alpha$ and $\gamma$ for the types and ideal chromosomes, respectively, were obtained after 20 iterations. The procedure follows the same protocol described in the human MiChroM work[8,40,41] that compares the probabilities of the simulated polymer with the experimental Hi-C map. The 3D structure representation of the chromosomes was built using Chimera and VMD software[94,95], and the coarse backbone representation was built using the Bendix VMD plugin[95,96]. Trajectory data, scripts for running MiChroM, and visualization of the 3D structures are available at the NDB server[19] (https://ndb.rice.edu). OpenMiChroM version 1.0.5 with CNDBtools was used for data analysis with Python 3[9,40]. The experimental data used in this work was obtained from the study of B. J. Matthews et al.[74] that is publicly available at NCBI (National Center for Biotechnology Information). The BioProject accession number that includes the Hi-C maps is PRJNA318737. Hi-C datasets are available at GEO (Gene Expression Omnibus) GSE113256 and at the Juicebox platform[97] (http://aidenlab.org/juicebox). Hi-C maps dense matrices were extracted using JuicerTools[98] (https://github.com/aidenlab/juicer) with Knight-Ruiz (KR) matrix balancing algorithm[99]. Comparisons between different normalizations are presented in the Supplementary Information (Supplementary Fig. 5). The ATAC-seq data were obtained from the same study of B. J. Matthews et al.[74] with reference number PRJNA418406 and SRX code SRX3580386). The short reads were aligned to the reference genome AaegL5.0 (NCBI - GCF_002204515.2) using the Bowtie 2 software package[100]. Files conversion and peaks calling were performed using Samtools[101] and Macs3[102]. The ATAC-seq signal values were integrated over 100 kb segments over the whole genome. High-intensity ATAC-seq signals are selected on values belonging to the 95th percentile and higher. On the other hand, Low-intensity signals are based on the 5th percentile and lower.

### Reporting summary

Further information on research design is available in the Nature Portfolio Reporting Summary linked to this article.

## Data availability

The authors declare that the data supporting this manuscript's findings are present in the main text or the Supplementary Information. All the publicly available datasets used in the study is followed by the accessible links/accession-codes. Additional relevant data are available from the corresponding authors upon reasonable request. The BioProject accession number that includes the Hi-C maps is PRJNA318737. Hi-C datasets are available at GEO (Gene Expression Omnibus) GSE113256. The ATAC-seq can be accessed with reference number PRJNA418406 and SRX code SRX3580386. The Aedes aegypti reference genome used is AaegL5.0 with NCBI code GCF_002204515.2.

## Code availability

The simulation code using OpenMiChroM is available at https://github.com/junioreif/OpenMiChroM. The code documentation with details on how to perform the chromatin dynamics simulations is available at https://open-michrom.readthedocs.io. The GROMACS molecular dynamics package is available at the Nucleome Data Bank (ndb.rice.edu)[19]. Analysis codes are now part of the CNDBtools implemented into the OpenMiChroM platform[9,40].

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

## Acknowledgements

The authors thank Antonio B. Oliveira Junior, Sumitabha Brahmachari, and Ryan Cheng for many useful conversations during the development of this work and for all of their comments and suggestions. This research was supported by the Center for Theoretical Biological Physics sponsored by the NSF (Grants PHY-2019745 and PHY-2210291) and by the Welch Foundation (Grant C-1792). JNO is a Cancer Prevention and Research Institute of Texas (CPRIT) Scholar in Cancer Research. VGC was also funded by FAPESP (São Paulo State Research Foundation and Higher Education Personnel), and CAPES (Higher Education Personnel Improvement Coordination) Grants 2016/13998-8 and 2017/09662-7. Additional support to PGW was provided by the D.R. Bullard-Welch Chair at Rice University (Grant No. C-0016). In addition to CTBP support, E.L.A. was supported by the Welch Foundation (Q-1866), a McNair Medical Institute Scholar Award, an NIH Encyclopedia of DNA Elements Mapping Center Award (UM1HG009375), a US-Israel Binational Science Foundation Award (2019276), the Behavioral Plasticity Research Institute (NSF DBI-2021795), and an NIH CEGS (RM1HG011016-01A1). MDP is supported by the NIGMS of the National Institutes of Health under award number

R35GM146852. The content is solely the responsibility of the authors and does not necessarily represent the official views of the National Institutes of Health. We would like to thank AMD for the donation of critical hardware and support resources from its HPC Fund that made this work possible.

## Author contributions

V.G.C., O.D., and M.D.P. conceived the project with support from the other authors. V.G.C. developed the codes, performed the simulations and the data analysis. V.G.C., O.D., E.L.A, P.G.W, J.N.O., and M.D.P contributed with technical advice and ideas. All authors participated in the scientific discussion. V.G.C. and M.D.P. wrote the initial draft of the paper, and all authors edited and refined the paper. All authors have approved the manuscript.

## Competing interests

The authors declare no competing interests.
