## [Peer Review File · Nature Communications]

REVIEWER COMMENTS

Reviewer #1 (Remarks to the Author):

In this work, the authors study the 3D architecture of the mosquito *Aedes aegypti* genome by data-driven physical simulations. They used a polymer model for chromatin includes phase separation driving the compartmentalization of chromatin and the effects of motor activity. Their analysis about the mechanical properties of the genome reveals the highly sensitive to the deformation of the nuclei about the chromosomes of *Aedes aegypti*, which provides a possible physical mechanism linking mechanical cues to gene regulation.

It may be potentially interested, however, the authors did not do a good job in presenting the data (where did they come from and how were they obtained) and writing the manuscript. Some figures are incomplete (see below for details) and the manuscript is short of necessary information. For example, it is not very clear where did the "experiments" data come from, and from the manuscript I could not find anywhere how they performed the effect of deforming the shape of the nucleus. It is not suitable for publication at all for the current "incomplete" manuscript.

Page 7. Paragraph 2. "Figures 1C and 1D show the Pearson's correlation matrix for the experimental and simulated data, respectively." The Figure 1C and 1D provide in the manuscript are not Pearson's correlation matrix.

Page 8. Paragraph 1. "Figure 1F presents the ..." But there is no Figure 1F in the Figure 1.

Page 8. Paragraph 1. "... (Coarsened 3D structure – Figure 1G)" There is no Figure 1G in Figure 1.

The supplementary figures (S1 and S2) have no figure legends.

Page 15 "The model was trained usinf the data for chromosome 1 of the *Aedes aegypti* obtained from the Hi-C map45" I assume "usinf" should be "using".

Reviewer #2 (Remarks to the Author):

Contessoto and co-authors built a physical model of *Aedes aegypti* chromatin and employed the model to investigate properties of genome architecture in these species. Using biophysical approaches is essential to understand chromatin organization in the nucleus. There are several models describing chromatin in mammalian or drosophila nuclei, however *Aedes aegypti* is less studied object, thus this works gives us new insights about genome organization across tree of life. Authors proposed important hypotheses: 1) increased DNA extrusion by SMC complexes in *Aedes aegypti* resulting in increased compaction of chromatin; 2) impact of the elongated shapes of chromosomes on global compartmentalization; 3) different sensibility of mammalian and *Aedes aegypti* chromatin to changes in nuclear shape. These are significant results, which justify publication; however, in my opinion these results should be backed up by additional data analysis (see points below) before acceptance.

Major points:

The title. Liquid crystalline properties are not discussed, moreover, the term "liquid crystal" appears only once in the text. Authors should provide more detailed description of "liquid crystal" and why chromatin of *Aedes aegypti* can be considered as an example of liquid

crystal. Alternatively, the title should be revised.

Describing differences between mammalian and *Aedes aegypti* chromatin architecture or specific features of chromatin packaging in *Aedes aegypti*, authors should provide quantitative analysis of the corresponding features and representative illustrations. In particular, the following important observations should be confirmed by data analysis:

2. 1. "As previously mentioned, both sets of Hi-C maps show very distinct properties when compared to those observed in most mammalian cell lines. The main diagonal appears particularly wide, indicating the frequent formation of contacts between loci further away in genomic distance than what is commonly seen in mammals" - I suggest performing direct comparison with mammalian data.

2.2. "Besides these two diagonals, the frequency of intra-chromosomal contacts is greatly depleted with respect to what is seen in mammalian cells" - same, I suggest performing quantitative analysis of the intra-chromosomal contacts.

2.3. "...in the mosquito's chromosomes short-range genomic contacts are enhanced, long-range contacts are depleted;" - in addition to quantitative analysis confirming this point, authors should address the issue of Hi-C data normalization. For example, consider biologically meaningful increase of all short-distance contacts (i.e. due to compaction) without changing distances between faraway loci. This change of short-distance contacts would result in decreased number of all observed long-distance contacts, because the total sum of contacts of each locus is constant after Hi-C matrix balancing. Therefore, authors should explain why they assume that one of the two observed effects (short-range genomic contacts are enhanced, long-range contacts are depleted) is not due to contacts normalization.

2.4. "Again for *Aedes aegypti*, active chromatin still forms droplets;" - does this follow from models?

2.5. "...but, because of the increased condensation and the elongated shape of territories, these droplets are less dynamical and become less likely to fuse with similar droplets situated at a distant position along the chromosomal axis." - again, it is not clear whether this is author's hypothesis (i.e. part of discussion) or this conclusion is supported by data.

2.6. "In contrast with what is seen in mammals, for *Aedes aegypti* the loci diffuse within local compartments but do not mix with far away chromatin even when the distant chromatin carries similar epigenetic markings." - similarly, this statement is not supported by data analysis.

3. "Crucially, we find that both polarization and the shortening of the chromosomes are needed for the emergence of this doubled-over architecture, reflected in the features observed in the Hi-C maps." - is this confirmed by modelling or other data analysis?

4. "ATACseq signal identifies accessible DNA regions and correlates positively with their belonging to A compartments" - please provide correlation coefficient as well as representative examples of ATAC-seq and compartment tracks to support this statement.

5. Please thoroughly revise all figures. For example: "Figures 1C and 1D show the Pearson's correlation matrix for the experimental and simulated data, respectively" - but Fig. 1C shows "3D quenched representative structure of chromosome 1 at 100kb resolution." and Fig 1D shows "Contact probability as a function of genomic distance." I was not able to find Pearson's correlation matrix. Same problem with figures in supplementary, i.e. "so, differential extrusion appears a more likely explanation rather than increased extrusion per

se (reference45 and figure S1).” - but figure S1 shows “representative 3D structures of chromosome 1 from *Aedes aegypti* modeled at the nominal information theoretic temperature ($T = 1.0$)”, so I can not understand how it confirms this statement.

Minor comment:

“The model was trained usinf ...” - typo in “usinf”

Reviewer #3 (Remarks to the Author):

In this work, Contessoto et al. do a computational study of chromatin organization of *Aedes aegypti* mosquito genome based on the recently available experimental data (in ref 45). The authors investigate the folded nature of chromosomes and the clustering of the centromeres/telomeres. Authors also simulate the shape change of chromosomes by applying mechanical tension on the genome, examining its implications.

The question addressed in this paper is an interesting one. However, the paper is written without much care; the manuscript lacks the necessary details to convince a reader that the results are correct and relevant.

The concerns are listed below:

1) First of all, the model: Even though the model is published earlier, authors must give all details necessary to reproduce the paper, at least in the supplementary material. The model section is written badly.

1a) What are the parameter values used? The model has many parameters-- the FENE spring parameters (R_0 and k_b), the LJ parameter ϵ , the two parameters (k_a and θ_0) in the three-body angle potential E_{cut} , r_0 , etc. One has to specify all these parameter values explicitly. How are they justified?

1b) The justification for the first equation on page S6 (the U_{sc} term) is unclear. Unfortunately, the authors have given no number for any of the equations in the Supplementary! So I cannot even refer to equations using a number!

1c) There are many R_0 variables in different equations. Are they all the same?

1d)The lengthwise compaction is important, and it is discussed in the main text. It is mentioned that an ideal chromosome potential will take care of the compaction and is related to SMC complexes' activity (extrusion?). What is this potential? Write down the explicit form. What are the corresponding parameters relevant for this work? Citing to old work is not sufficient. In the supplementary, there is no mention of this potential at all.

1e)What is in table S2?

1f) How is the force (tension) introduced in the model? I could not find even the word "tension" or "force" anywhere in the model section!

Now, about the results:

2) The figure-1 is completely messed up!

2a) Even though authors talk about Figure 1F and Figure 1G in the text, there are no 1F and 1G figures!

2c) "Figures 1C and 1D show the Pearson's correlation matrix for the experimental and simulated data, respectively" -- This is not true! Fig 1C is a 3D configuration. There is no matrix!

2d) "Pearsons' correlation between the simulation and the experiments is $R = 0.957$ ". Please have the plot showing the high correlation, experimental data in the X-axis, and simulation data in the Y-axis.

2e) Pearsons correlation as a function of genomic distance (in SI): the correlation between experiment and simulation is often 0.5 or smaller (often close to zero). What does this imply?

3) About the contact probability map, 3D shape, etc:

3a) In Figure-1 A and B, the scale bar has no quantitative number. What are the "high" and "low" values? It is not clear how the contact probabilities are normalized.

3b) Since the contact probability values are relative, what is crucial is the 3D (spatial) distance (distance between different regions of the genome). To ensure that the predictions are sensible, one has to compare 3D distance between some segments with an experimentally measured 3D distance value. At least, the authors must show that the R_g value obtained from the simulation is comparable to the physical size of the chromosomes.

3c) What is the size of a bead in actual units (nanometer)? What is the size of the chromosomes in nanometers or micrometers?

Many earlier papers systematically compare the simulation results with 3D distance, discuss the normalization, etc. For example, Giorgetti et al., Cell 2014 (Cell 157, 950–963 2014) and their other papers; Chiariello... Nicodemi, Cell reports 2020 (and their other papers), Kumari et al. Biophys. J. 118, 2193–2208 (2020), Brackley et al. Genome Biology 17:59, (2016). Guang Shi et al. Nat Com, 2018, PRX 2021. None of these papers (except Shi et al. 2018.) are cited. Authors must cite them and discuss the 3D distance/size of the chromosome. Only then any "physical" modeling will be meaningful.

4) Applying tension

4a) How is the tension applied? From the figure description, it appears that the authors are working in a constant extension ensemble (the extension value is specified). Is this true? How is this implemented? What is the corresponding force/pressure?

4b) Is the tension applied on two specific beads? (Telomere/centromere beads?) or on a set of beads?

5) Other comments:

5a) How are mosquito parameters different from other parameters (human/yeast)? It will be good to have a table comparing the parameters.

5b) Since centromere and telomere clustering are known in other organisms (like yeast), it will make complete sense if you compare the results with other organisms.

5c) Title talks about "liquid crystalline" nature. How is liquid crystalline order characterized? Use standard order parameters used in the liquid crystal literature.

5d) In figure 2, there is mention of 1M and 1P. In the SI, there are C1 and C2. Explain all the symbols in respective captions.

Reply to the reviewers for the manuscript entitled: “**Interphase chromosomes of the *Aedes aegypti* mosquito are liquid crystalline and can sense mechanical cues**”.

We want to thank the reviewers for the careful reading and analysis of the manuscript and for providing thoughtful comments, including several suggestions on how to improve this work. Below, we address all questions and suggestions, with the original comments reported in *italics*. **Corrections and additions in the revised manuscript are marked in red color.** Please find below our responses and a description of the revisions based on the comments of the reviewers:

Reviewer #1:

*In this work, the authors study the 3D architecture of the mosquito *Aedes aegypti* genome by data-driven physical simulations. They used a polymer model for chromatin includes phase separation driving the compartmentalization of chromatin and the effects of motor activity. Their analysis about the mechanical properties of the genome reveals the highly sensitive to the deformation of the nuclei about the chromosomes of *Aedes aegypti*, which provides a possible physical mechanism linking mechanical cues to gene regulation.*

It may be potentially interested, however, the authors did not do a good job in presenting the data (where did they come from and how were they obtained) and writing the manuscript. Some figures are incomplete (see below for details) and the manuscript is short of necessary information. For example, it is not very clear where did the “experiments” data come from, and from the manuscript I could not find anywhere how they performed the effect of deforming the shape of the nucleus. It is not suitable for publication at all for the current “incomplete” manuscript.

Author's reply: Thank you for your comment. We included in the main text additional information regarding the experimental data used in this work. The text contains information about the Hi-C maps and ATAC-seq experiments availability and accession numbers:

The experimental data used in this work was obtained from the study of B. J. Matthews et al. that is publicly available at NCBI (National Center for Biotechnology Information). The BioProject accession number that includes the Hi-C maps is PRJNA318737. Hi-C datasets are valuable at GEO (Gene Expression Omnibus) GSE113256 and at the Juicebox platform (<http://aidenlab.org/juicebox>). Hi-C maps dense matrices were extracted using

JuicerTools (<https://github.com/aidenlab/juicer>) with Knight-Ruiz (KR) matrix balancing algorithm.

The ATAC-seq data were obtained from the same study of B. J. Matthews et al. with reference number PRJNA418406 and SRX code SRX3580386. The short reads were aligned to the reference genome AaegL5.0 (NCBI - GCF_002204515.2) using the Bowtie 2 software package. Files conversion and peaks calling were performed using Samtools and Macs3. The ATAC-seq signal values were integrated over 100kb segments over the whole genome. High-intensity ATAC-seq signals are selected on values belonging to the 75th percentile and higher. On the other hand, Low-intensity signals are based on the 25th percentile and lower.

Page 7. Paragraph 2. “Figures 1C and 1D show the Pearson’s correlation matrix for the experimental and simulated data, respectively.” The Figure 1C and 1D provide in the manuscript are not Pearson’s correlation matrix.

Page 8. Paragraph 1. “Figure 1F presents the ...” But there is no Figure 1F in the Figure 1.

Page 8. Paragraph 1. “... (Coarsened 3D structure – Figure 1G)” There is no Figure 1G in Figure 1. The supplementary figures (S1 and S2) have no figure legends.

Author's reply: Thank you for your comment. We apologize for the confusion created due to the mislabeled figure 1 and the lack of information in the main text and SI. Figure 1 panels, legends, and discussion are now corrected in the revised form of the manuscript. We added more detailed legends for figures S1 and S2 in the Supplementary material.

Page 15 “The model was trained usinf the data for chromosome 1 of the Aedes aegypti obtained from the Hi-C map45” I assume “usinf” should be “using”.

Author's reply: Thank you for pointing this out. The typo is not corrected in the revised form of the manuscript.

Reviewer #2:

Contessoto and co-authors built a physical model of Aedes aegypti chromatin and employed the model to investigate properties of genome architecture in these species. Using biophysical approaches is essential to understand chromatin organization in the nucleus. There are several models describing chromatin in mammalian or drosophila nuclei, however Aedes aegypti is less studied object, thus this works gives us new insights about genome organization across tree of life. Authors proposed important hypotheses: 1) increased DNA extrusion by SMC complexes in Aedes aegypti resulting in increased compaction of chromatin; 2) impact of the elongated shapes of chromosomes on global compartmentalization; 3) different sensibility of mammalian and Aedes aegypti chromatin to changes in nuclear shape. These are significant results, which justify publication; however, in my opinion these results should be backed up by additional data analysis (see points below) before acceptance.

Major points:

The title. Liquid crystalline properties are not discussed, moreover, the term “liquid crystal” appears only once in the text. Authors should provide more detailed description of “liquid crystal” and why chromatin of Aedes aegypti can be considered as an example of liquid crystal. Alternatively, the title should be revised.

Author's reply: Thank you for the appropriate request. We expanded our discussion on the liquid crystalline properties of the Aedes genome. To better explain the partially condensed structures observed in our simulations, we included a new analysis that shows the formation of hierarchical layers and local structures of the 3D structures. This discussion is included in the revised form of the manuscript and presented in the text parts below:

Introduction:

Compartmentalization is observed along the two diagonals, indicating that chromatin in *Aedes aegypti*, while partially condensed by strong short-range length-wise compaction, remains fluid and can rearrange to accommodate phase separation⁴¹. It is worthwhile mentioning that liquid crystalline structures have also been suggested as the genomic architecture for some systems⁶¹⁻⁶³.

Results:

As mentioned before, such genome architecture resembles that of liquid crystals (partially ordered). The chromatin fiber of *Aedes aegypti* flows and changes shapes similar to a human chromosome in interphase that is described as being

liquid-like. This first aspect, like liquid droplets, allows for A/B phase separation and compartmentalization (observed in the anti-diagonal in figures 1A and 1B). However, there is local structural order in the form of helices that leads to an orientational order along the genomic distance that is similar to what is seen for the mitotic chromosome - thicker diagonal in Hi-C maps of figures 1A and 1B). To quantify such structural property, we also calculated an orientation order parameter (O_{OP}) previously employed to investigate local rearrangements in mitotic chromosomes⁴¹. The parameter O_{OP} is defined as the correlation of a unit vector of a bead i and a bead $i+4$ (see methods section for details). The analyses were performed for the quenched structure and the ensemble of 3D structures at $T=1.0$ (see SI Figures S1E and S1F for details). Figure 1F shows that O_{OP} oscillates as a function of the genomic separation that can be associated with fibril structures⁴¹. The Fourier transform of O_{OP} presents two regions with intense values of the spectrum, indicating that there are two layers of fibril structures as shown in figure 1G. The first is related to a higher turn frequency with a periodicity around 0.6 Mb which can be observed as local helicoidal structures shown in figure 1C. In addition, the second layer has a periodicity of longer genomic separation in the range of 8 Mb.

Describing differences between mammalian and Aedes aegypti chromatin architecture or specific features of chromatin packaging in Aedes aegypti, authors should provide quantitative analysis of the corresponding features and representative illustrations. In particular, the following important observations should be confirmed by data analysis:

2. 1. *“As previously mentioned, both sets of Hi-C maps show very distinct properties when compared to those observed in most mammalian cell lines. The main diagonal appears particularly wide, indicating the frequent formation of contacts between loci further away in genomic distance than what is commonly seen in mammals” - I suggest performing direct comparison with mammalian data.*

2.2. *“Besides these two diagonals, the frequency of intra-chromosomal contacts is greatly depleted with respect to what is seen in mammalian cells” - same, I suggest performing quantitative analysis of the intra-chromosomal contacts.*

Author's reply: We agree that these points need clarification and quantitative analysis; thank you for suggesting this. In our recent work, *Hoencamp C., et al. Science (2021)*, we performed in situ Hi-C on 24 species, and the architectural features observed in those maps can be divided into two groups, type-I and type-II. Type-I group (that

includes the *Aedes aegypti*) incorporates Rabl-like configurations such as centromere clustering, telomere clustering, and telomere-to-centromere axis. Type-II group (human and other mammals) includes the genome organization related to chromosome territories. These architectural features are identified using a scoring parameter called ACA (Aggregate Chromosome Analysis). ACA aggregates the signal from all intra- and inter-chromosomal contacts, and the scores are defined as observed-to-expected ratios (*Hoencamp C., et al. Science (2021) - SI*). The ACA chromosome territory score $S(ct)$ for the *Aedes aegypti* is 1.109 which is low when compared to 11.150 from the Human genome or 6.179 from the Tammar wallaby chromosomes (mammalian). These ACA values defined the Human and Tammar wallaby genomes to belong to the type-II group which forms territories. On the other hand, the low value of $S(ct)$ includes the mosquito in group type-I.

In addition to this ACA score discussion, we compare the contact probability as a function of the genomic distance for chromosome 1 from *Aedes* and humans (Figure S3). The curves have similar decay until the region around 2 Mb where the mosquito genome shows higher probabilities than the human until it reaches the values close to 11 Mb. The difference in the decay in this length scale is related to higher compaction of the *Aedes* genome. On the other hand, for genomic separation longer than 11 Mb, the human chromosome shows a higher contact frequency in comparison to the mosquito which suggests the formation of long-range interaction and the formation of territories. This discussion is now included in the Support Information and in the main text:

Results:

In our recent work⁴³, we quantified these properties by calculating the ACA (Aggregate Chromosome Analysis) on in situ Hi-C maps of 24 species. The architectural features observed in those maps can be divided into two groups, type-I (Rabl-like configurations such as centromere clustering, telomere clustering, and telomere-to-centromere axis) and type-II (chromosome territories). The details of the ACA score and the comparison of the contact probability curves for the human and mosquito chromosome 1 are presented in the SI.

2.3. “...in the mosquito’s chromosomes short-range genomic contacts are enhanced, long-range contacts are depleted;” - in addition to quantitative analysis confirming this point, authors should

address the issue of Hi-C data normalization. For example, consider biologically meaningful increase of all short-distance contacts (i.e. due to compaction) without changing distances between faraway loci. This change of short-distance contacts would result in decreased number of all observed long-distance contacts, because the total sum of contacts of each locus is constant after Hi-C matrix balancing. Therefore, authors should explain why they assume that one of the two observed effects (short-range genomic contacts are enhanced, long-range contacts are depleted) is not due to contacts normalization.

Author's reply: Thank you for your question. Indeed, Hi-C data normalization can over or underestimate the frequency of contacts performed by a locus. The Hi-C map used in the model training was balanced using the KR - Knight-Ruiz algorithm that is implemented in Juicer software package (N. C. Durand et al., Cell Systems, 2016). In order to address the possible artifacts generated by the normalization, we calculated the contact frequency as a function of the genomic distance for the *Aedes aegypti* chromosome 1 using four different normalization methods available in Juicer. The curve decay and a discussion text are now included in the SI (Figure S4). The scaling curves do not present significant variations when comparing different normalization methods. We can then assume that there is no over- or underestimation of short- or long-range contacts due to matrix balancing.

2.4. *“Again for Aedes aegypti, active chromatin still forms droplets;” - does this follow from models?*

2.5. *“...but, because of the increased condensation and the elongated shape of territories, these droplets are less dynamical and become less likely to fuse with similar droplets situated at a distant position along the chromosomal axis.” - again, it is not clear whether this is author's hypothesis (i.e. part of discussion) or this conclusion is supported by data.*

2.6. *“In contrast with what is seen in mammals, for Aedes aegypti the loci diffuse within local compartments but do not mix with far away chromatin even when the distant chromatin carries similar epigenetic markings.” - similarly, this statement is not supported by data analysis.*

Author's reply: These are good observations. The formation of micro phase-separation between A/B compartments is reported in the first study that introduces the Minimal Chromatin Model (MiChroM) that is employed here for the mosquito genome. Given the mosquito genome present compartmentalization off-diagonal, we expect that the A/B phase-separation is occurring in a similar way to that observed in human chromosomes. However, due to the Rabl-like genome architecture of the mosquito, contacts between

loci that are located near the polarized regions (Centromere and Telomeres) are less frequently formed. We modified the text to clarify these points in the revised version of the manuscript:

Results:

Nevertheless, significant differences between the chromosomal architecture of mammals and *Aedes* mosquitoes are seen even in the case of active chromatin positioning. In mammals, we have observed that the chromatin segments belonging to the same genomic compartment - and thus carrying similar epigenetic markings - form liquid droplets. In three dimensions, these droplets rearrange dynamically by splitting and fusing, leading to the emergence of genome-wide compartments observed in DNA-DNA ligation assays^{8,18,64}. In *Aedes aegypti*, similarly to what one finds for human chromosomes, the active chromatin forms droplets (A/B micro phase-separation); but, due to the increased condensation and the elongated shape of the territories, these droplets are less likely to fuse with similar droplets situated at distant positions along the chromosomal axis. Thus, the global structure leads to the formation of only local compartments. In contrast with what is seen in mammals, in *Aedes aegypti* the loci diffuse only within local compartments but do not mix with far away chromatin even when the distant chromatin carries similar epigenetic markings to a given local compartment, *i.e.*, the Rabl-like genome architecture of mosquitoes, causes the contacts between loci that are located near the polarized regions (Centromere and Telomeres) to form less frequently. The distinct nature of compartments in *Aedes* is likely to have some repercussions on transcriptional regulation.

3. *“Crucially, we find that both polarization and the shortening of the chromosomes are needed for the emergence of this doubled-over architecture, reflected in the features observed in the Hi-C maps.” - is this confirmed by modelling or other data analysis?*

Author's reply: These observations come from the simulations. When we performed the simulation of the mosquito chromosome 1 but using the parameters trained on the human genome, we observed a folded and spherical shape of the chromosome even when applying the centromere and telomeres polarization restraint. This suggests that

polarization is not enough for generating a Rabl-like genome architecture but also a particular balance of motor activity and phase separation that leads to a partially condensed genome.

4. "ATACseq signal identifies accessible DNA regions and correlates positively with their belonging to A compartments" - please provide correlation coefficient as well as representative examples of ATAC-seq and compartment tracks to support this statement.

Author's reply: Thank you for your comment. In this work, we use the first component of the eigenvectors extracted from the correlation matrix of the experimental Hi-C map to determine the A/B compartments. The ATAC-seq signals are divided into two groups based on the signal intensity, High and Low. The group denominated High contains signal values above the 95th percentile. On the other hand, the group Low corresponds to signal values below the 5th percentile. The signal distribution with the percentile cuts is presented in FigureS5 with the eigenvector AB distribution and an explanation text. There are a total of 156 elements in each percentile cut. From the High group, 100 loci are identified belonging to the A compartment which corresponds to 64% agreement of a locus being classified as A type and having a high signal value of the ATAC-seq track. In addition, there are 91 loci identified as B compartments that give 59% hits. We reformulated the text to discuss this topic in detail:

Results:

The ATAC-seq signal identifies accessible DNA regions and it has been reported to correlate positively with their belonging to A compartments^{83,84}. For *Aedes aegypti*, we observe that the high-intensity values of the ATAC-seq signal, in the majority (64%), belong to the A compartments, while the B compartments contain most of the low-intensity signal (59%) values (see Figure S5 and SI for details). Thus, high-intensity values allow the compartment to be identified as an A compartment despite the fact that Hi-C experiments were performed using whole body extract, while the ATAC-seq experiments were performed using only brain cells.

5. Please thoroughly revise all figures. For example: “Figures 1C and 1D show the Pearson’s correlation matrix for the experimental and simulated data, respectively” - but Fig. 1C shows “3D quenched representative structure of chromosome 1 at 100kb resolution.” and Fig 1D shows “Contact probability as a function of genomic distance.”. I was not able to find Pearson’s correlation matrix. Same problem with figures in supplementary, i.e. “so, differential extrusion appears a more likely explanation rather than increased extrusion per se (reference45 and figure S1).” - but figure S1 shows “representative 3D structures of chromosome 1 from *Aedes aegypti* modeled at the nominal information theoretic temperature ($T = 1.0$)”, so I can not understand how it confirms this statement.

Author's reply: Thank you for your comment. This is a similar comment of reviewer #1; we apologize for the confusion created due to the mislabeled figure 1 and the lack of information in the main text and SI. Figure 1 panels, legends, and discussion are now corrected in the revised form of the manuscript. We added more detailed legends for figures S1 and S2 in the Supplementary material.

Minor comment:

“The model was trained usinf ...” - typo in “usinf”

Author's reply: Thank you for pointing this out. The typo is not corrected in the revised form of the manuscript.

Reviewer #3:

*In this work, Contessoto et al. do a computational study of chromatin organization of *Aedes aegypti* mosquito genome based on the recently available experimental data (in ref 45). The authors investigate the folded nature of chromosomes and the clustering of the centromeres/telomeres. Authors also simulate the shape change of chromosomes by applying mechanical tension on the genome, examining its implications.*

The question addressed in this paper is an interesting one. However, the paper is written without much care; the manuscript lacks the necessary details to convince a reader that the results are correct and relevant.

The concerns are listed below:

1) *First of all, the model: Even though the model is published earlier, authors must give all details necessary to reproduce the paper, at least in the supplementary material. The model section is written badly.*

1a) *What are the parameter values used? The model has many parameters-- the FENE spring parameters (R_0 and k_b), the LJ parameter epsilon, the two parameters (k_a and θ_0) in the three-body angle potential E_{cut} , r_0 , etc. One has to specify all these parameter values explicitly. How are they justified?*

Author's reply: Thank you for your comment. We agree that there is a lack of details regarding the model implementation and parameters. We modified the Support Information text to present the parameter values explicitly. We chose a set of parameters that have been extensively employed in several studies of polymer physics and genome modeling (refs 8,9,18,19,37,38,40-44,66,75,77,93 in the main paper). The parameters $R_0=1.5\sigma$, $k_b=30\epsilon/\sigma^2$ and $k_a=2\epsilon$ are reported in the literature as a set of parameters that avoid bond crossing and allow for a time step in the simulation compared to a fluid, where $\epsilon = K_B T$ and $\sigma=1$ in reduced units. That means, increasing k_b would require a reduction of the simulation time step. $E_{cut} = 4\epsilon$ is a finite energetic cost allowing for chain crossing (topoisomerases activity).

1b) *The justification for the first equation on page S6 (the U_{sc} term) is unclear. Unfortunately, the authors have given no number for any of the equations in the Supplementary! So I cannot even refer to equations using a number!*

Author's reply: Thank you for your comment. The equation describing the potential U_{sc} is assigned as number (5) in the support information. U_{sc} is a soft-core repulsive potential that is applied to a pair of non-bonded beads that mimics the activity of topoisomerases by allowing for chain crossing given a finite energetic cost of $E_{cut} = 4\epsilon$

1c) *There are many R_0 variables in different equations. Are they all the same?*

Author's reply: We apologize for the confusion. R_0 is the equilibrium bond length in the FENE bond potential (equation 2). R_{f_0} (equation 6) is the center flat bottom potential location applied for restraint centromere and telomeres in the nucleus wall.

1d) The lengthwise compaction is important, and it is discussed in the main text. It is mentioned that an ideal chromosome potential will take care of the compaction and is related to SMC complexes' activity (extrusion?). What is this potential? Write down the explicit form. What are the corresponding parameters relevant for this work? Citing to old work is not sufficient. In the supplementary, there is no mention of this potential at all.

Author's reply: Thank you for your comment. We fit the data point resulting from the optimization procedure employing a stretched exponential function. The parameter values, the data points, and the curve fit are presented in Figure S6A in the SI.

1e) What is in table S2?

Author's reply: This table, now labeled as table S3, presents the interaction strength for different chromatin types A and B. These parameter values are obtained after the optimization procedure. We include an explanation text in the SI.

1f) How is the force (tension) introduced in the model? I could not find even the word "tension" or "force" anywhere in the model section!

Author's reply: Thank you for your comment. The force/tension is related to the position restraint applied to centromeres and telomeres to maintain the polarized architecture, assuming their interaction with the nucleus membrane. Increasing/decreasing the centromere-telomeres spatial distance would mimic the effect of the nucleus deformation by expansion and compression. We include an explanation in the SI when introducing the restraint potential.

Now, about the results:

2) The figure-1 is completely messed up!

2a) Even though authors talk about Figure 1F and Figure 1G in the text, there are no 1F and 1G figures!

2c) "Figures 1C and 1D show the Pearson's correlation matrix for the experimental and simulated data, respectively" -- This is not true! Fig 1C is a 3D configuration. There is no matrix!

Author's reply: Thank you for your comment. This is a similar observation made by reviewers 1 and 2; we apologize for the confusion created due to the mislabeled Figure

1. Figure 1 panels, legends, and discussion are now corrected in the revised form of the manuscript.

2d) *"Pearsons' correlation between the simulation and the experiments is $R = 0.957$ ". Please have the plot showing the high correlation, experimental data in the X-axis, and simulation data in the Y-axis.*

2e) *Pearsons correlation as a function of genomic distance (in SI): the correlation between experiment and simulation is often 0.5 or smaller (often close to zero). What does this imply?*

Author's reply: The overall Pearson's correlation value obtained when comparison simulations and experimental Hi-C maps is mostly dictated by the scaling curve (polymer compaction). The scatter plot comparing simulations and experiments is presented in Figure S6B with the curve fit values of the slope, intercept, standard deviation, and correlation coefficient. The Pearson's correlation as a function of genomic distance curves presents the improvement of the AB-only model in comparison to a homopolymer or random data set (baseline).

3) *About the contact probability map, 3D shape, etc:*

3a) *In Figure-1 A and B, the scale bar has no quantitative number. What are the "high" and "low" values? It is not clear how the contact probabilities are normalized.*

Author's reply: Thank you for your comment. The experimental Hi-C is normalized using the KR - Knight-Ruiz algorithm that is implemented in Juicer software package (N. C. Durand et al., Cell Systems, 2016). High and Low values correspond to probability 1 and 0, respectively. These values are now included in the received version of Figure 1.

3b) *Since the contact probability values are relative, what is crucial is the 3D (spatial) distance (distance between different regions of the genome). To ensure that the predictions are sensible, one has to compare 3D distance between some segments with an experimentally measured 3D distance value. At least, the authors must show that the R_g value obtained from the simulation is comparable to the physical size of the chromosomes.*

3c) *What is the size of a bead in actual units (nanometer)? What is the size of the chromosomes in nanometers or micrometers?*

Many earlier papers systematically compare the simulation results with 3D distance, discuss the normalization, etc. For example, Giorgetti et al., Cell 2014 (Cell 157, 950–963 2014) and their other papers; Chiariello... Nicodemi, Cell reports 2020 (and their other papers), Kumari et al. Biophys. J. 118, 2193–2208 (2020), Brackley et al. Genome Biology 17:59, (2016). Guang Shi et al. Nat Com, 2018, PRX 2021. None of these papers (except Shi et al. 2018.) are cited.

Authors must cite them and discuss the 3D distance/size of the chromosome. Only then any "physical" modeling will be meaningful.

Author's reply: That is a good point. In our previous works, we employed a bead diameter of $\sigma_{50\text{kb}} = 0.165\mu\text{m}$ using the MiChroM (Minimal Chromatin Model) energy function at 50 kb resolution. Here we used a locus resolution of 100 kb per model bead. Assuming a constant density of chromatin between different bead resolutions, the bead diameter for the 100 kb resolution is $\sigma_{100\text{kb}} = \sqrt[3]{2} \sigma_{50\text{kb}} = 0.20789\mu\text{m}$. This value leads to the estimate of the centromere-telomeres distance (which corresponds to the nucleus diameter $38\sigma_{100\text{kb}}$) around $7.9\mu\text{m}$. The total length of *Aedes aegypti* chromosome 1 in interphase is approximately $15.3\mu\text{m}$. It is reported in the literature that chromosome 1 in the earlier stage of mitosis has an average length of $11.86\mu\text{m}$ (Timoshevskiy et al., PNTD,2013). This is quite good agreement given that there should be some condensation during the initial stages of mitosis which is related to chromosome shortening. We added this discussion in the revised form of the manuscript and included the physical modeling references.

4) Applying tension

4a) How is the tension applied? From the figure description, it appears that the authors are working in a constant extension ensemble (the extension value is specified). Is this true? How is this implemented? What is the corresponding force/pressure?

4b) Is the tension applied on two specific beads? (Telomere/centromere beads?) or on a set of beads?

Author's reply: These are good questions. Yes, we are working in a constant extension ensemble. We add the details on how to create the initial state in the SI - model section. We perform temperature annealing (from theoretical temperature $T=2.0$ to $T=1.0$) to randomize the initial state, to avoid possible kinetic traps and intermediate state due to centromere-telomeres constraints. We employed a position restraint where a group of beads are spatially constraint by a flat-bottomed potential harmonic potential (equation 6 - SI). The group of beads corresponding to the centromere and telomeres move freely within the restraint radius ($R_{f0} = 2.5\sigma$)

5) *Other comments:*

5a) *How are mosquito parameters different from other parameters (human/yeast)? It will be good to have a table comparing the parameters.*

5b) *Since centromere and telomere clustering are known in other organisms (like yeast), it will make complete sense if you compare the results with other organisms.*

Author's reply: In this work, we modeled the mosquito genome at a resolution of 100 kb per bead/locus. A comparison with other organisms/models can be performed in terms of similar physical observations such as motor activity, phase separation between different chromatin types, and centromere-telomeres polarized architecture. We include in the SI a comparison of the mosquito with the human parameter obtained in the first paper of MiChroM energy function at 50 kb.

5c) *Title talks about "liquid crystalline" nature. How is liquid crystalline order characterized? Use standard order parameters used in the liquid crystal literature.*

Author's reply: Thank you for the appropriate request. This is a similar comment from reviewer 2. We expanded our discussion on the liquid crystalline properties of the *Aedes* genome. To better explain the partially condensed structures observed in our simulations, we included a new analysis that shows the formation of hierarchical layers and local structures of the 3D structures. This discussion is included in the revised form of the manuscript and presented in the text parts below:

Introduction:

Compartmentalization is observed along the two diagonals, indicating that chromatin in *Aedes aegypti*, while partially condensed by strong short-range length-wise compaction, remains fluid and can rearrange to accommodate phase separation⁴¹. It is worthwhile mentioning that liquid crystalline structures have also been suggested as the genomic architecture for some systems⁶¹⁻⁶³.

Results:

As mentioned before, such genome architecture resembles that of liquid crystals (partially ordered). The chromatin fiber of *Aedes aegypti* flows and changes shapes similar to a human chromosome in interphase that is described as being liquid-like. This first aspect, like liquid droplets, allows for A/B phase separation

and compartmentalization (observed in the anti-diagonal in figures 1A and 1B). However, there is local structural order in the form of helices that leads to an orientational order along the genomic distance that is similar to what is seen for the mitotic chromosome - thicker diagonal in Hi-C maps of figures 1A and 1B). To quantify such structural property, we also calculated an orientation order parameter (O_{OP}) previously employed to investigate local rearrangements in mitotic chromosomes⁴¹. The parameter O_{OP} is defined as the correlation of a unit vector of a bead i and a bead $i+4$ (see methods section for details). The analyses were performed for the quenched structure and the ensemble of 3D structures at $T=1.0$ (see SI Figures S1E and S1F for details). Figure 1F shows that O_{OP} oscillates as a function of the genomic separation that can be associated with fibril structures⁴¹. The Fourier transform of O_{OP} presents two regions with intense values of the spectrum, indicating that there are two layers of fibril structures as shown in figure 1G. The first is related to a higher turn frequency with a periodicity around 0.6 Mb which can be observed as local helicoidal structures shown in figure 1C. In addition, the second layer has a periodicity of longer genomic separation in the range of 8 Mb.

5d) *In figure 2, there is mention of 1M and 1P. In the SI, there are C1 and C2. Explain all the symbols in respective captions.*

Author's reply: Thank you for pointing this out. The labels 1M and 1P are used for distinguishing between the two copies of the same chromosome when simulating the full nucleus. C1 is employed when the analysis/data are obtained for the chromosome without distinguishing between the homologous chromosomes. We added a text explaining the labels' differences and updated the figures.

REVIEWERS' COMMENTS

Reviewer #1 (Remarks to the Author):

With more necessary information added, I read this version of the manuscript with more interest, and this work might provide important insight for the evolution of genome organization. However, the authors need to make more modifications to support publication.

The authors answered some of my questions and corrected the figures and manuscript, but they still did not answer and explain "how they performed the effect of deforming the shape of the nucleus". This part is still very confusing.

I saw Reviewer 3 also asked "How is the force (tension) introduced in the model? ", the authors answered in the rebuttal letter, however did not put the answers into the revised manuscript.

The SI information should properly be introduced in the manuscript, not just supplied as a separate file without introducing each of the methods and figures.

"Model and Simulation Details" are still not detailed enough.

The author's answer to the typo on Page 15 "usinf" still had a typo --- "The typo is not corrected in the revised form of the manuscript." Or the authors insist not to change?

Reviewer #2 (Remarks to the Author):

The authors addressed most of my comment, but there are several minor issues to be resolved:

1 (related to Q2.3). Contacts normalization. While authors have confirmed their observations with 4 different matrix balancing methods, all these methods equalize number of contacts per loci. From this follows that any of them can not discriminate increase of short-range interactions and decrease of long-range interactions. Authors should discuss this possible explanation of the observed data along with their hypotheses ("...in the mosquito's chromosomes short-range genomic contacts are enhanced, long-range contacts are depleted; so, differential extrusion appears a more likely explanation rather than increased extrusion per se.")

2 (related to Q4). ATAC-seq analysis.

"For *Aedes aegypti*, we observe that the high-intensity values of the ATAC-seq signal, in the majority (64%), belong to the A compartments, while the B compartments contain most of the low-intensity signal (59%) values (see Figure S5 and SI for details)."

While revised analysis indeed shows statistical enrichment of high signal ATAC-peaks in A-compartment and low-signal ATAC peaks in B-compartment, this enrichment is moderate (64% compared to 50% expected; 59% compared to 50% expected). No statistical analysis was done (i.e. by shuffling E1 values) to confirm that this enrichment is statistically significant. For example, 99% confidence interval of success probability derived from empirical Bernoulli distribution with $n_{\text{trials}}=156$ and $n_{\text{success}}=91$ is $0.47716 \leq p \leq 0.68429$. And even if the result is statistically significant, authors should clearly state that there is disagreement between compartments and ATAC-peaks for many loci, which may affect downstream analysis and conclusions.

3. "Table S1: Kolmogorov-Smirnov statistic two-sided test (KS-value) comparing ... The distributions are presented in Figure __4D__ of the main text". - I assume authors referred to Fig. 3D

4. Both elongated forms of the chromosomes and its impact on compartmentalization were previously

suggested for *Drosophila* ([https://doi.org/10.1016/S0960-9822\(99\)80509-0](https://doi.org/10.1016/S0960-9822(99)80509-0)) and more recently for *Anopheles* mosquitoes (<https://doi.org/10.1038/s41467-022-29599-5>). Authors may consider noting this in discussion.

Reviewer #3 (Remarks to the Author):

The authors have addressed most of my concerns. I have only a few additional comments.

1) The sentence "Oop is defined as the correlation of a unit vector of a bead i and a bead $i + 4$ " does not make sense. However, the definition given in eq.8 in the SI does make sense. Therefore, the sentence should be rewritten to convey what is defined via the equation. Maybe something like this:

Correlation between two unit vectors connecting beads $(i, i+4)$ and $(j, j+4)$...

2) If you can provide an equivalent of Figure 1G for human chromosomes, that would be a good control to demonstrate what is special about this genome.

3) In Figure 1A, 1B scale bar, please also give some numbers between 0 and 1. The way it is shown, a reader would have no idea whether it is shown on a log scale or a linear scale.

Reply to the reviewers for the manuscript entitled: “**Interphase chromosomes of the *Aedes aegypti* mosquito are liquid crystalline and can sense mechanical cues**”.

We want to thank the reviewers for the careful reading and analysis of the manuscript and for providing thoughtful comments, including several suggestions on how to improve this work. Below, we address all questions and suggestions, with the original comments reported in *italics*. **Corrections and additions in the revised manuscript are marked in red color.** Please find below our responses and a description of the revisions based on the comments of the reviewers:

Reviewer #1:

With more necessary information added, I read this version of the manuscript with more interest, and this work might provide important insight for the evolution of genome organization. However, the authors need to make more modifications to support publication.

The authors answered some of my questions and corrected the figures and manuscript, but they still did not answer and explain “how they performed the effect of deforming the shape of the nucleus”. This part is still very confusing.

I saw Reviewer 3 also asked “How is the force (tension) introduced in the model? ”, the authors answered in the rebuttal letter, however did not put the answers into the revised manuscript.

The SI information should properly introduced in the manuscript, not just supplied as a separate file without introducing each of the method and figures.

“Model and Simulation Details” are still not detailed enough.

The author’s answer to the typo on Page 15 “usinf” still had a typo --- “The typo is not corrected in the revised form of the manuscript.” Or the authors insist not to change?

Author's reply: Thank you again for your comments. Besides the section “Model and Simulation Details” (Now named ‘Methods’) in the main text, the section “Method Details” in the SI explains in detail all the potential employed in the simulation protocol.

The nucleus deformation is performed by increasing or decreasing the centromere-telomeres (CT) spatial distance. The CT positions are spatially restrained to consider the effect of their anchoring to the nucleus wall. Variations in the CT distance would result in an expansion or compression of the genome. We also included the section “Centromere and Telomeres Position Restraints” in the SI, which describes how tension is applied.

The typo presented on page 15 is now corrected. Thank you for your comment

Reviewer #2:

The authors addressed most of my comment, but there are several minor issues to be resolved: 1 (related to Q2.3). Contacts normalization. While authors have confirmed their observations with 4 different matrix balancing methods, all these methods equalize number of contacts per loci. From this follows that any of them can not discriminate increase of short-range interactions and decrease of long-range interactions. Authors should discuss this possible explanation of the observed data along with their hypotheses ("...in the mosquito's chromosomes short-range genomic contacts are enhanced, long-range contacts are depleted; so, differential extrusion appears a more likely explanation rather than increased extrusion per se.")

Author's reply: We have added in SI a comparison between the different normalization schemes and raw (un-normalized) Hi-C matrices of *Aedes*. The analysis shows that the normalization does not introduce any artifact in the observed behavior. We thank the reviewer for the comment, as the new analysis does indeed clarify and strengthen our findings.

2 (related to Q4). ATAC-seq analysis.

*"For *Aedes aegypti*, we observe that the high-intensity values of the ATAC-seq signal, in the majority (64%), belong to the A compartments, while the B compartments contain most of the low-intensity signal (59%) values (see Figure S5 and SI for details)."*

While revised analysis indeed shows statistical enrichment of high signal ATAC-peaks in A-compartment and low-signal ATAC peaks in B-compartment, this enrichment is moderate (64% compared to 50% expected; 59% compared to 50% expected). No statistical analysis was done (i.e. by shuffling E1 values) to confirm that this enrichment is statistically significant. For example, 99% confidence interval of success probability derived from empirical Bernoulli distribution with $n_{\text{trials}}=156$ and $n_{\text{success}}=91$ is $0.47716 \leq p \leq 0.68429$. And even if the result is statistically significant, authors should clearly state that there is disagreement between compartments and ATAC-peaks for many loci, which may affect downstream analysis and conclusions.

Author's reply: Thank you for the appropriate comment. The reviewer is right. In fact, the correlation values between compartments A/B and the high/low-intensity signal of the ATAC-seq were not the goal of our analysis. The language used was misleading. We have changed the text to make our findings clearer.

Results:

*For *Aedes aegypti*, we do indeed find a mild correlation, with the high-intensity values of the ATAC-seq signal belonging in the majority (64%) to the A*

compartments, while the B compartments contain most of the low-intensity signal (59%) values (see Figure S5 and SI for details). The low correlation is possibly due to the fact that Hi-C experiments were performed using whole body extract, while the ATAC-seq experiments were performed using only brain cells.

3. "Table S1: Kolmogorov-Smirnov statistic two-sided test (KS-value) comparing ... The distributions are presented in Figure __4D__ of the main text". - I assume authors referred to Fig. 3D

Author's reply: Thank you for pointing this out. The figure indication is not corrected in the revised form of the manuscript.

4. Both elongated forms of the chromosomes and its impact on compartmentalization were previously suggested for *Drosophila* ([https://doi.org/10.1016/S0960-9822\(99\)80509-0](https://doi.org/10.1016/S0960-9822(99)80509-0)) and more recently for *Anopheles mosquitos* (<https://doi.org/10.1038/s41467-022-29599-5>). Authors may consider noting this in discussion.

Author's reply: Thank you for your comment. The *Drosophila* work is cited in the manuscript as ref. 86 (Wilkie G S. et al., 1999). The *Anopheles mosquitos* reference was cited as the bioRxiv version in ref. 85. We have now updated this reference with the published version (Lukyanchikova, V. et al, 2022).

Reviewer #3:

The authors have addressed most of my concerns. I have only a few additional comments.

1) The sentence "Oop is defined as the correlation of a unit vector of a bead i and a bead $i + 4$ " does not make sense. However, the definition given in eq.8 in the SI does make sense. Therefore, the sentence should be rewritten to convey what is defined via the equation. Maybe something like this:

Correlation between two unit vectors connecting beads ($i, i+4$) and (j and $j+4$)...

Author's reply: Thank you for your suggestion. We modified the text and the SI to better explain the parameter O_{OP} .

Results:

The parameter O_{OP} is defined as the correlation between two unit vectors connecting beads $[i, i+4]$ and $[j, j+4]$ (see SI for details).

2) *If you can provide an equivalent of Figure 1G for human chromosomes, that would be a good control to demonstrate what is special about this genome.*

Author's reply: Thank you for your suggestion. We included the calculation of the orientation order parameter and its Fourier transform in the SI Figure S2.

3) *In Figure 1A, 1B scale bar, please also give some numbers between 0 and 1. The way it is shown, a reader would have no idea whether it is shown on a log scale or a linear scale.*

Author's reply: Thank you for your comment. We included the scale bar in all figures of the revised form of the text.